# Serological and Molecular Characterization of Hepatitis C Virus-Related Cryoglobulinemic Vasculitis in Patients without Cryoprecipitate

**DOI:** 10.3390/ijms241411602

**Published:** 2023-07-18

**Authors:** Cecilia Napodano, Gabriele Ciasca, Patrizia Chiusolo, Krizia Pocino, Laura Gragnani, Annunziata Stefanile, Francesca Gulli, Serena Lorini, Gessica Minnella, Federica Fosso, Riccardo Di Santo, Sabrina Romanò, Valerio Basile, Valerio De Stefano, Gian Ludovico Rapaccini, Anna Linda Zignego, Enrico Di Stasio, Mariapaola Marino, Umberto Basile

**Affiliations:** 1Department of Laboratory Medicine and Pathology, S. Agostino Estense Hospital, 41126 Modena, Italy; cecilia.napodano@gmail.com; 2Sezione di Fisica, Dipartimento di Neuroscienze, Università Cattolica del Sacro Cuore, 00168 Rome, Italy; gabriele.ciasca@unicatt.it (G.C.); riccardo.disanto92@gmail.com (R.D.S.); sabrina.romano@unicatt.it (S.R.); 3Fondazione Policlinico Universitario “A. Gemelli” I.R.C.C.S., 00168 Rome, Italy; patrizia.chiusolo@unicatt.it (P.C.); gessica.minnella@policlinicogemelli.it (G.M.); federica.fosso@policlinicogemelli.it (F.F.); valerio.destefano@policlinicogemelli.it (V.D.S.); gianludovico.rapaccini@unicatt.it (G.L.R.); 4Sezione di Ematologia, Dipartimento di Scienze Radiologiche ed Ematologiche, Università Cattolica del Sacro Cuore, 00168 Roma, Italy; 5Unità Operativa Complessa di Patologia Clinica, Ospedale Generale di Zona San Pietro Fatebenefratelli, 00189 Rome, Italy; krizia.pocino@gmail.com (K.P.); stefanile.nunzia@gmail.com (A.S.); 6Department of Translation Research and New Technologies in Medicine and Surgery, Università di Pisa, 56126 Pisa, Italy; laura.gragnani@unipi.it; 7Unit of Clinical Pathology, Bambino Gesù Children’s Hospital I.R.C.C.S., 00165 Rome, Italy; francesca.gulli@opbg.net; 8Department of Experimental and Clinical Medicine, Interdepartmental Centre MASVE, University of Florence, 50121 Florence, Italy; serena.lorini@unifi.it (S.L.); annalinda.zignego@unifi.it (A.L.Z.); 9Clinical Pathology Unit and Cancer Biobank, Department of Research and Advanced Technologies, I.R.C.C.S. Regina Elena National Cancer Institute, 00144 Rome, Italy; valeriobasile90@gmail.com; 10Dipartimento di Scienze Biotecnologiche di Base, Cliniche Intensivologiche e Perioperatorie, Università Cattolica del Sacro Cuore, 00168 Rome, Italy; enrico.distasio@unicatt.it; 11Sezione di Patologia Generale, Dipartimento di Medicina e Chirurgia Traslazionale, Università Cattolica del Sacro Cuore, 00168 Rome, Italy; 12Dipartimento di Patologia Clinica, Ospedale Santa Maria Goretti, A.U.S.L. Latina, 04100 Latina, Italy; u.basile@ausl.latina.it

**Keywords:** HCV, cryoglobulins, mixed cryoglobulinemia, IGH, TCR, FLC, RF, IgG, IgM

## Abstract

Prolonged B cells stimulation due to the Hepatitis C virus (HCV) can result in autoimmunity, stigmatized by rising levels of cryoglobulins (CGs), the rheumatoid factor (RF), and free light chains (FLC) of immunoglobulins (Ig) associated with a range of symptoms, from their absence to severe cryoglobulinemic vasculitis and lymphoma. Here, we aimed to identify an immunological signature for the earliest stages of vasculitis when cryoprecipitate is still not detectable. We firstly analyzed the IgG subclasses, FLC, and RF in 120 HCV-RNA-positive patients divided into four groups according to the type of cryoprecipitate and symptoms: 30 asymptomatic without cryoprecipitate (No Cryo), 30 with vasculitis symptoms but without CGs that we supposed were circulating but still not detectable (Circulating), 30 type II and 30 type III mixed cryoglobulinemia (Cryo II and Cryo III, respectively). Our results revealed that patients with supposed circulating CGs displayed a pattern of serological parameters that closely resembled Cryo II and Cryo III, with a stronger similarity to Cryo II. Accordingly, we analyzed the groups of Circulating and Cryo II for their immunoglobulin heavy chain (IgH) and T-cell receptor (TCR) gene rearrangements, finding a similar mixed distribution of monoclonal, oligoclonal, and polyclonal responses compared to a control group of ten HCV-RNA-negative patients recovered from infection, who displayed a 100% polyclonal response. Our results strengthened the hypothesis that circulating CGs are the origin of symptoms in HCV-RNA-positive patients without cryoprecipitate and demonstrated that an analysis of clonal IGH and TCR rearrangements is the best option for the early diagnosis of extrahepatic complications.

## 1. Introduction

HCV is a hepatotropic and lymphotropic virus capable of establishing a chronic infection, leading to hepatocellular carcinoma and extrahepatic disorders such as mixed cryoglobulinemia (MC) [1]. Cryoglobulinemia is defined by the presence of cryoglobulins (CGs), cryoprecipitable immune complexes clinically characterized by a small-vessel vasculitis involving mainly the skin, joints, peripheral nervous system, and the kidneys. A reduction in temperature is believed to trigger some steric modification of molecules with the exposition of non-polar residues, reduction of solubility, and cryoprecipitation. When the temperature is restored to 37 °C, these “unidentified molecules” circulating in the blood revert to their initial conformation. Type I cryoglobulinemia, which consists of only one isotype or subclass of monoclonal immunoglobulin (Ig), is frequently asymptomatic. On the other hand, type II and type III cryoglobulinemia, classified as mixed cryoglobulinemia because they include IgG and IgM, are often detected in the course of infectious and systemic diseases. In this scenario, essential MC represents a distinct entity strikingly associated with HCV infection (>90%) [1] characterized by leukocytoclastic vasculitis of small- and medium-sized vessels, with cutaneous and multiple visceral organ involvement. Type II and III cryoglobulins are immune complexes composed of polyclonal IgGs, i.e., autoantigens, and mono- or polyclonal IgM, respectively. IgM are the corresponding autoantibodies to rheumatoid factor (RF) activity. A wide range of genetic, environmental, and immunological factors may contribute to the development of MC, which is clinically characterized by a classical triad of purpura, weakness, and arthralgia [1,2].

A direct-acting antiviral usually leads to the improvement/remission of cryoglobulinemic vasculitis, although symptoms may persist/recur after a sustained virological response (SVR) [3]. The HCV has developed multiple strategies to evade host immune surveillance, enabling the establishment of persistent infections in most infected individuals [4]. This persistent infection in the host may account for the wide variety of both autoimmune and lymphoproliferative disorders in HCV-infected individuals. HCV-induced B-cell dysregulation is probably the result of an indirect process arising from the chronic antigenic stimulation of a limited pool of preexisting autoreactive B cells; the prolonged stimulation of the HCV is necessary for abnormal B-cell lymphoproliferation, as eradication of the HCV typically results in the resolution of both HCV-related MC and non-Hodgkin’s lymphoma (NHL) [5,6]. In this setting, monitoring well-defined immunological biomarkers of B-cell activation as IgG subclasses, free light chains (FLCs) of Ig, C3 and C4 complement fractions, RF IgM, and RF IgG appear to be useful in detecting and eventually preventing the “escape” of a B-cell clone [7,8,9,10,11,12].

Although representing the main cause of MC [1], HCV infection may often be asymptomatic, and 19% of patients are aware of their hepatitis status randomly through routine blood tests. The molecular mechanisms underlying the different evolutionary pathways, as well as the biomarkers of a specific disease state, are not yet fully understood [7,8,13]. Some HCV-infected patients develop cryoglobulinemia, but not all of them exhibit symptoms; conversely, some patients show typical MC symptoms but lack detectable cryoglobulins, which represent the hallmark of the disease. It has been postulated that, in these patients, CGs may circulate in the serum of these symptomatic individuals but still not be detectable due to the early stage of the disease [14]. Different studies suggest that viral infections trigger specific pathogenetic pathways leading to the development of lymphoproliferative disorders. However, these pathways appear to be influenced by the genetic backgrounds of the affected individuals [15,16,17]. Various single-nucleotide polymorphisms (SNPs) located on different genes, including the BAFF cytokine system on chromosome 6, have been identified as potential early markers for malignancy development [3].

B- and T-cell cooperation is essential to sustain an effective immune response; direct and indirect evidence has demonstrated that B-cell depletion may impair T-cell activation, and, on the other hand, the dysregulation of this crosstalk has profound effects on the elicit of autoimmunity [18]. A molecular assay is the method employed for an immunogenetic analysis, mostly concerning clonal rearrangements of genes of immunoglobulin (IG) and T-cell receptor (TCR) loci. The acquisition of specific receptors clonally distributed is an essential step in the differentiation pathway of B and T cells, allowing the production of a repertoire of unique antigen receptors presenting in all clonal progeny, referred to as B- and T-cell clonality. Through the recombination of clonal-specific immunoreceptor gene rearrangements, a polyclonal repertoire of IG/TR (immunoglobulin/T-cell receptor) receptors is created. In certain autoimmune diseases, this repertoire is oligoclonal, whereas, in lymphoid malignancies, receptors are largely monoclonal [19,20]. IG/TR rearrangements thus represent unique genetic biomarkers (molecular signatures) for studying immune cells for clinical, diagnostic, and research applications [21]. The further determination of an immunoglobulin heavy chain locus (IGH) somatic mutation is the marker identification and quantification of the minimal residual disease in lymphoid neoplasms [22].

Since the pathogenesis of chronic HCV-related MC involves multiple steps, we investigated various serological biomarkers of B-cell activation, including IgG subclasses, FLCs, RF IgG, and RF IgM, and performed a molecular analysis of clonal rearrangements of immunoglobulin and T-cell receptor genes. The study was designed for two different phases consisting of an initial analysis of serological biomarkers in the entire population of 120 HCV-RNA-positive patients divided into four groups and, secondly, an evaluation of specific rearrangements of TCR/IGH genes in selected groups of patients, with the addition of a control group of 10 patients who were negative for HCV viremia. The steps of the analysis are visualized in Figure 1, and the baseline correlates of the 120 HCV-RNA-positive patients enrolled are summarized in Table 1. We analyzed the serological and molecular features of HCV-RNA-positive patients who did not have detectable cryoglobulins (referred to as “circulating” patients) but exhibited symptoms of MC in comparison to HCV-RNA-positive patients without vasculitis symptoms or cryoglobulins (asymptomatic No Cryo), as well as to HCV-RNA-positive patients with type II and type III CGs (Cryo II and Cryo III, respectively). Our objective was to identify the immunological signature of the earliest stages of the disease.

## 2. Results 

### 2.1. General Features of HCV-RNA-Positive Patients

The characteristics of all the patients are graphically visualized in Figure 2, depicting the distribution of sex (Figure 2A), HCV genotype (Figure 2B), cryocrit levels (Figure 2C), and CG-related symptoms (Figure 2D). No statistically significant differences were observed in age (*p* = 0.37) and sex (*p* = 0.75) among the groups. The studied cohort primarily consisted of genotypes I and II, which aligned with the expected distribution in an Italian population of HCV-positive individuals [23].

Packed CGs were observed only in patients with type II (Cryo II) and type III (Cryo III) CGs after centrifugation at 4 °C (Figure 2C). The levels of cryocrit significantly differed between these two groups (*p* = 0.0077). HCV patients without CGs did not exhibit any related symptoms, while patients with presumed circulating CGs exhibited a symptom pattern that closely resembled the Cryo II and Cryo III patients, leaning more towards the Cryo II pattern (Figure 2D).

### 2.2. Comparative Analysis of Serological Correlates of HCV-RNA-Positive Patients

In Table 2, we summarize the serological parameters and viremia levels of the HCV-RNA-positive patients. In Figure 3, we show a heatmap depicting the levels of all the measured biomarkers (heatmap rows) across each subject (heatmap columns) to capture the interindividual variability.

In Figure 4, we show the distribution of IgG subclasses in the four groups. A black dashed line representing the upper reference level for each investigated parameter is superimposed on the graph. Irrespective of the patient group, we observed a prominent prevalence of IgG1, followed by IgG2 and IgG3, while IgG4 appeared relatively less abundant.

Interestingly, higher average levels of IgG1 and IgG3 were observed in patients with circulating CGs and Cryo II, while the remaining groups exhibited lower levels. Consistently, the average levels of IgG1 exceeded the reference threshold only in patients with circulating CGs, whereas those with Cryo II were near the threshold. Similarly, for IgG3, the average values were consistent with the upper threshold within one standard deviation only in the aforementioned two groups.

As indicated in Table 2, the levels of IgG1 (*p* = 0.006) and IgG3 (*p* < 0.001) were significantly higher in the group with circulating CGs compared to the asymptomatic subjects without CGs. Additionally, significant differences in IgG3 levels were observed among the patients with Cryo II, Cryo III, and supposed circulating CGs compared to the asymptomatic patients (No Cryo). There was a relatively less notable, yet statistically significant, difference (*p* = 0.041) in the IgG2 levels between patients with circulating CGs and those with Cryo II. No statistically significant differences were observed in the IgG4 levels among the four groups.

In Figure 5, we show the FLC serum levels and the k/λ ratio in the different groups. A black dashed line representing the upper reference level for each investigated parameter is superimposed on the graph. The average FLC-k levels exceeded the reference range in each group, while only the average λ levels in the patients with Cryo II exceeded the threshold, along with the patients with presumed circulating CGs, which approached it. Notably, only the patients with Cryo II and circulating CGs exceeded the k/λ ratio. In all three parameters, the patients with Cryo and circulating CGs displayed higher values, while the remaining groups exhibited comparatively decreased values. Overall, the FLCs displayed a similar pattern to what we observed in IgG1 and IgG3, with the patients with presumed circulating CGs closely resembling those with Cryo II (Table 2).

We compared the levels of the RF IgG and RF IgM among the four investigated groups (Figure 6). Once again, a similar pattern to the IgGs and FLCs was observed, with the RF IgM and RF IgG generally being higher in patients with Type II and circulating CGs compared to the remaining groups. RF IgM was significantly increased in Cryo II compared to Cryo III (*p* = 0.001) and No Cryo (*p* < 0.001). RF IgG appeared to be significantly elevated in patients with circulating CGs compared to type III (*p* < 0.001) and No Cryo (*p* < 0.001).

For the sake of completeness, in Figure 7, we analyzed the correlation among the immunological parameters in the four different groups of HCV-RNA-positive patients. The correlations were arranged as a Pearson’s coefficient matrix. Correlations highlighted with the “x” symbol were not statistically significant, according to the power analysis. It could be noticed that the inflammatory IgG and FLC responses were highly correlated in the patients without CGs. However, this correlation was lost in the patients with CGs. Additionally, in the asymptomatic patients without CGs, RF IgG and viremia displayed several correlations with other serological parameters. Overall, the group of asymptomatic patients displayed more systematic and strong correlations among the serological parameters, while the remaining groups exhibited fewer and sparse correlations. Interestingly, despite the presence of a statistically significant difference in viremia between Cryo II and the other groups (Table 2), no correlation was observed between viremia and the other parameters in patients with Cryo II. Taken together, the results of Figure 7 suggest that the group of patients with presumed circulating CGs clinically resembled the Cryo II and Cryo III groups, behaving differently from the group of patients without CGs, which displayed a higher degree of correlations among the serological parameters.

### 2.3. Analysis of TCR and IGH Clonal Rearrangements

In Figure 8, we compared the clonal rearrangements of TCRγ and IGH FR1, FR2, and FR3 of 10 HCV-RNA-negative patients without CGs (left) who were resolved from acute infection with the group of 30 HCV-RNA-positive patients with presumed circulating CGs (center) and with HCV-RNA-positive patients with Cryo II (right). Significant differences were observed among the three groups for all the measured variables in terms of their monoclonal, oligoclonal, and polyclonal responses. A qualitative observation of Figure 8 reveals that HCV-RNA-negative patients without CGs displayed a 100% polyclonal response. A different scenario was observed in HCV-RNA-positive patients with presumed circulating CGs and Cryo II, as both groups exhibited mixed behavior in terms of their monoclonal, oligoclonal, and polyclonal responses. This finding further emphasized the similarity between the two groups and strengthened the hypothesis of the presence of circulating CGs in patients displaying MC-related symptoms but without a detectable cryoprecipitate.

The results are summarized in Table 3, with a statistical analysis conducted using Fisher’s exact test and adjusted with Bonferroni correction. Significant differences were observed in the clonality frequencies of TCR-A (*p* = 0.023), TCR-B (*p* = 0.01), IGH-FR2 (*p* = 0.003), IGH-FR2 (*p* < 0.001), and IGH-FR3 (*p* = 0.009). Notably, a post hoc analysis carried out using Bonferroni correction for the *p*-values (Table 3) confirmed that the observed difference was only between patients without CGs and the remaining groups. In contrast, no significant differences were observed between Cryo II and patients with possible circulating CGs, further highlighting the clinical similarities between these two groups, already observed in the clinical symptoms (Figure 2D), IgGs (Figure 4), FLCs (Figure 5), and RF (Figure 6).

## 3. Discussion

### 3.1. B-Cell Proliferation and Clonal Selection in HCV-Related Mixed Cryoglobulinemia

Mixed cryoglobulinemia is a common extrahepatic manifestation of HCV infection characterized by the intravascular deposition of immune complexes, including RF IgM/IgG, polyclonal IgG, and viral RNA, leading to inflammation [24]. The proliferation of B cells plays a critical role in the development of extrahepatic manifestations and lymphoproliferative complications associated with MC. Furthermore, genetic factors contribute to the increased risk of HCV-related lymphoproliferative disorders, as specific SNPs on various genes have been associated with this condition [3,15,25].

Even after viral clearance, patients with HCV-related MC exhibit the survival of an aberrant B-cell clone, which explains the persistence of immune abnormalities and clinical symptoms. This dysregulation undergoes a progressive and multiphase clonal selection process, resulting in the production of a specific type of oligoclonal IgG subclass with RF activity. In some cases, this lymphoproliferative condition can progress to B-cell NHL [24,26,27].

### 3.2. Symptomatic Similarity in HCV-Positive Patients with Circulating and Type II Cryoglobulins

The cryocrit level is the primary biomarker used to diagnose MC patients. However, in some HCV-infected individuals, typical MC symptoms may arise without a detectable cryoprecipitate. The underlying mechanics of this phenomenon are not yet fully understood, leading to challenges in diagnosing and managing these patients.

In this study, our objective was to investigate the hypothesis presented in the literature that CGs circulate in the serum of these patients but remain undetectable during the early stages of the disease [14]. To address this issue, we conducted a study involving 120 HCV-RNA-positive patients, categorized into four distinct groups. Group 1 comprised asymptomatic patients, while Group 2 consisted of symptomatic patients without detectable cryoprecipitate. Furthermore, Group 3 and Group 4 included patients with type II (Cryo II) and type III (Cryo II) CGs, respectively. Notably, all patients in Groups 2, 3, and 4 exhibited symptoms of MC.

Upon analyzing the baseline clinical characteristics of the different patient groups (Figure 2D), we observed that symptomatic patients without detectable cryoprecipitate (Group 2) displayed similar clinical symptoms to Cryo II and Cryo III, setting them apart from asymptomatic patients (Group 1). Interestingly, when examining the frequency of these symptoms, Group 2 showed an intermediate profile. They had a higher frequency of symptoms compared to Cryo III, and their symptom frequency was similar but slightly lower compared to Cryo II.

These findings highlight the need for a distinct clinical pathway for the individuals in Group 2, considering them the most likely as patients with MC, probably type II. This hypothesis was further investigated in our manuscript, as described in subsequent sections. Moreover, these observations support the hypothesis that there may be circulating CGs that are not detectable as cryoprecipitates but still contribute to the symptomatic manifestations experienced by these patients.

### 3.3. Beyond Clinical Symptoms: Similarities in Serological Parameters Highlight the Relationship between Patients with Circulating and Type II Cryoglobulins

To better characterize the patients in Group 2 who exhibited typical MC symptoms but lacked the characteristic cryoprecipitate for a definitive diagnosis, we analyzed a set of widely used serological parameters (Table 2). To achieve this, we quantified the levels of the IgG subclasses, recognizing their crucial role in the development of the disease [28,29]. Specifically, the profiles of IgG subclasses can vary depending on the type and duration of antigen exposure, a concept known as “subclass restriction” [28]. This suggests that similarities in the IgG profiles might underline a common origin of the pathology. In this regard, previous findings have shown that MC patients exhibit elevated levels of the IgG3 subclass, along with the clonal expansion of B cells producing RF IgG [26].

In agreement with the literature, we observed distinct differences in the IgG3 levels among Cryo II, Cryo III, and presumed circulating GCs compared to asymptomatic individuals. Very interestingly for our purposes, a close resemblance in marker levels was observed between Cryo II and circulating CGs, both falling within one standard deviation of the maximum threshold for this parameter (Figure 4). A similar pathway was also observed for the IgG1 subclass, which displayed comparable levels in the circulating and Cryo II groups (Figure 4). Of note, the average level of IgG1 exceeded the normal threshold only in patients with circulating CGs; the IgG1 levels of Cryo II closely approached the threshold, while the other classes showed significantly lower average values.

The observation of elevated levels not only of IgG3 but also of IgG1 in both groups is interesting and warrants further discussion. It is known that IgG3 monoclonal CGs with RF activity can induce significant extrahepatic manifestations. However, its short half-life may restrict the duration of an abnormal inflammatory response. This raises the hypothesis that the risk of clonal expansion is higher when cryoglobulins are initially formed by two IgG subclasses [26,29], a condition that might be compatible with what we observed in Figure 4 in the case of IgG1 and IgG3.

In various immunological disorders, including MC, abnormal activation of the immune system can result in the increased production and activities of plasma cells, leading to an increase in circulating FLCs, which serve as useful biomarkers in a wide range of immunopathological conditions [30,31,32]. Currently, in clinical practice, the measurement of FLCs assumes that their unbalanced production during a monoclonal gammopathy determines an alteration of their ratio (k/λ FLC), a sensitive and specific marker of clonality [33]. Patients with HCV-related MC and NHL indeed exhibit an altered FLC ratio (k/λ) and significantly higher FLC levels compared to healthy subjects [7,8]. Moreover, emerging evidence suggests that FLCs, previously regarded as metabolic byproducts, possess distinct biochemical and structural characteristics, potentially endowing them with biological functions, including the activation of the immune system. Thus, in patients with HCV-related MC, FLCs can potentially play a role in the process of CG formation by stimulating B cells in a site-specific manner.

Due to the potential relevance of FLCs as biomarkers and triggers of MC, we investigated the FLC levels (k-FLC and λ-FLC) and the k/λ ratio in the four patient groups (Figure 5). In agreement with the discussed literature, we observed increased levels of k-FLC for all the subjects investigated, with the average k levels exceeding the upper limit of the normal range. Very interestingly, like what we observed in IgGs, for all the FLC-related parameters, patients with circulating CGs closely resembled those with type II CGs, while the other groups displayed reduced values. Remarkably, the average k/λ ratio exceeded the upper limit of the normal range only in the circulating and Cryo II groups, highlighting how this could be considered a potential clinical parameter for achieving a better classification for the former group.

In Figure 6, we investigated the levels of RF IgM and RF IgG, as these autoantibodies are crucial components of CG complexes that form and deposit into various tissues, ultimately contributing to the clinical manifestations of MC. Interestingly, once again, a closely resembling pathway was revealed: the levels of IgM/IgG-RF were found to be increased in patients with type II and circulating CGs, while the remaining patient groups exhibited lower values.

Taken together, the findings presented in Figure 4, Figure 5 and Figure 6 demonstrate an intriguing similarity in terms of the serological parameters between patients with presumed circulating CGs and those with type II CGs, systematically distinguishing these two groups from asymptomatic individuals. The comparable values of these markers and determinants of MC, namely IgGs, FLCs, and RF, not only help explain the observed similarities in the symptoms highlighted in Figure 2D but also provide further evidence that, despite the absence of detectable cryoprecipitate, these patients represent a distinct clinical population compared to asymptomatic individuals. Taking this into consideration, these results suggest that the discussed parameters can be used to improve the laboratory characterization of patients exhibiting typical MC symptoms but who have not received a diagnosis due to the absence of a cryoprecipitate.

### 3.4. Comparative Analysis of Clonal Rearrangements Reveals Similar Patterns in Patients with Circulating and Type II Cryoglobulins

In the second part of our study, we focused on HCV-RNA-positive patients with detectable type II cryoprecipitate (Group 3) and those with presumed circulating CGs (Group 2); a control group of 10 HCV-RNA-negative patients without CGs or symptoms was included for comparison (Figure 1). We aimed to investigate more in-depth the previously discussed similarity between the Cryo II and circulating groups in terms of clinical symptoms (Figure 2D) and the serological laboratory parameters (Figure 4, Figure 5 and Figure 6). The analysis aimed to investigate the clonal rearrangements of TCR and IGH genes to elucidate the variations in monoclonal, oligoclonal, and polyclonal responses among the three study groups. The results, as depicted in Figure 8, demonstrated significant differences in clonality patterns among the groups. Specifically, the control group consisting of HCV-recovered patients exhibited a 100% polyclonal response, indicating a diverse and non-clonal B-cell population. In contrast, the HCV-positive groups with type II cryoglobulins and presumed CGs displayed similar clonality patterns, characterized by varying frequencies of clonal expansions. The rationale for examining both TCR and IGH genes lies in the complex interplay between T and B lymphocytes. While the primary focus was on IGH genes (encoding B-cell receptors), it is important to consider the critical role of T-cell activation in modulating B-cell responses. Therefore, the analysis also encompassed the evaluation of TCR genes to obtain a comprehensive understanding of the clonality process in mixed cryoglobulinemia. Consistent with previous studies [14] and our own findings, MC can manifest in an incomplete form where the cryoprecipitate of serum CGs is undetectable. In these cases, we hypothesize that clonality patterns may serve as a valuable screening tool to identify patients with chronic HCV infection who harbor circulating CGs that have not yet precipitated. This is particularly relevant when these patients present with clinical symptoms resembling those of MC.

The presence of clonal rearrangement in both the circulating and Cryo II groups supports the claim that, in the former, CGs at low concentrations do not reach the necessary cluster size to overcome the solubility limit and form a precipitate. In this regard, understanding the cellular and molecular events that induce alterations in the solubility of steric molecules is crucial for comprehending the wide range of pathophysiological mechanisms associated with this phenomenon.

## 4. Materials and Methods

### 4.1. Patients

We studied 120 HCV-RNA-positive patients, enrolled at two Italian centers (Center for Systemic Manifestations of Hepatitis Viruses (MaSVE), Department of Experimental and Clinical Medicine, University of Florence, Florence, Italy, and Fondazione Policlinico Agostino Gemelli-I.R.C.C.S. Università Cattolica Del Sacro Cuore, Rome, Italy). The presence of MC symptoms was recorded according to the classification criteria for MC as proposed by the Italian Group for the Study of Cryoglobulinemias in 1989 and later revised in 2002 [14].

The study was conceived through two different steps. The first one consisted of an evaluation of B-cell activity persistence, analyzing serum biomarkers of the whole 120 HCV-positive patients characterized by the presence of HCV-RNA in the serum, and by the absence of antiviral treatment and/or immunosuppressive therapy, subdivided into 4 groups according to symptoms and CGs:

Group 1 (No Cryo): Thirty HCV-positive patients who were asymptomatic (no MC symptoms or CGs).

Group 2 (Circulating): Thirty HCV-positive patients who displayed symptoms referred to Meltzer’s triad syndrome (purpura, arthralgia, and asthenia), suggesting the diagnosis of cryoglobulinemia but without a cryoprecipitate (we supposed that CGs were circulating in these patients but still undetectable because they were at the very early stage of the disease).

Group 3 (Cryo II): Thirty HCV-positive patients who displayed MC symptoms and type II CGs.

Group 4 (Cryo III): Thirty HCV-positive patients who displayed MC symptoms and type III CGs.

The second step was an immunogenetic analysis concerning clonal rearrangements of IGH and TCR, analyzing Groups 2 (HCV-positive, MC symptoms, postulated circulating CGs) and 3 (HCV-positive, MC symptoms, type II CGs) in comparison with a control group of 10 patients recovered from HCV infection who were negative for HCV-RNA but positive for anti-HCV antibodies without MC symptoms (Figure 1).

Exclusion criteria: coinfection with HIV or HBV, the presence of severe comorbidities not related to cryoglobulinemic vasculitis (i.e., no hepatic or hematologic malignancies, including NHL), and the concomitant administration of therapies for vasculitis (i.e., rituximab and plasma exchange cycles). Patients who previously underwent these treatments (at least 6 months prior), as well as patients taking low-medium doses of corticosteroids (0.1–0.5 mg/kg/day) or symptomatic drugs/measures (i.e., colchicine/low antigen diet), were included with the presence of cryofibrinogen.

All patients enrolled had their liver stiffness measured by transient elastography, which correlated well with the fibrosis METAVIR stages. Transient elastography is advantageous, performed repeatedly, does not require a highly experienced operator, and has a low risk of complications. The main demographic, clinical, and virological characteristics of the HCV-positive population are reported in Table 1.

The whole study was conducted in accordance with the Declaration of Helsinki and approved by the Ethical Committee at the Università Cattolica del Sacro Cuore (Immuno-HVR; ID: 2080); all the participants provided written informed consent prior to enrollment. All samples were processed anonymously.

### 4.2. Laboratory Testing

Sera were obtained by standard centrifugation, divided into aliquots, and stored frozen until analysis. Samples were thawed only once and immediately assayed in a blinded fashion and in a single batch. FLCs were assessed using the Freelite™ Human Kappa and Lambda Free Kits (The Binding Site, Birmingham, UK) using an Optilite instrument (The Binding Site, Birmingham, UK; free k normal range: 3.3–19.4 mg/L; free λ normal range: 5.7–26.3 mg/L). A ratio of k/λ < 0.26 or > 1.65 was abnormal, according to the manufacturer’s recommendations [34].

The four IgG subclasses were measured by turbidimetry through the employment of Human IgG and IgG subclass liquid reagent kits (The Binding Site, Birmingham, UK) with an Optilite instrument according to the manufacturer’s recommendations. These kits were performed for quantifying human IgG and IgG subclasses. The concentrations were automatically calculated by reference to a standard curve stored within the instrument. Normal ranges for the subclasses were 3.82–9.29 g/L for IgG1, 2.42–7.0 g/L for IgG2, 0.22–1.76 g/L for IgG3, and 0.04–0.86 g/L for IgG4. Samples were tested according to the manufacturer’s instructions, and serum dilutions, where necessary, were performed according to the manufacturer’s recommendations.

RF IgG and RF IgM were tested using ELISA kits RF-IgG/IgM (Menarini, Florence, Italy). All samples were analyzed at the same time following the manufacturer’s instructions, and plates were immediately read using a plate reader, as indicated by the manufacturer.

A sample of 10 cc was collected and kept at 37 °C in prewarmed tubes without an anticoagulant for ≥1 h (until complete clotting). Serum was separated, transferred to Wintrobe tubes, and stored at 4 °C [35]. The presence of precipitate in these samples was determined by visual inspection before centrifugation and after 7 days of cold incubation (4 °C). Samples were centrifuged at 2500× *g* for 10 min at 4 °C, and the amount of CGs was estimated as cryocrit, the percentage of the volume occupied by the precipitated proteins compared with the total volume of serum.

A plasma sample of 10 cc was collected in a citrate tube and kept at 37 °C to evaluate the presence of cryofibrinogen. Centrifugation at 3500 rpm for 10 min was performed at 37 °C. After the plasma (for cryofibrinogen detection) was decanted into two tubes, it was kept at 4 °C for 72 h.

For the analysis of the clonal rearrangement of IGH and TCRγ, the DNA was extracted by human peripheral blood mononuclear cells using the QIAmp DNA Blood Mini Kit^®^.

For the detection of the T-cell receptor γ chain gene, the “T-CELL Lymphoma Kit FL” (Experteam, Venice, Italy) was used, which identified the clonal rearrangements of the TCRγ chain by means of 2 semi nested PCR reactions (A and B). Consensus primers covering the Vγ1–Vγ9 segments were used in both reactions, while consensus primers covering JGT1/2 and JGT3 were used in the first and second reactions, respectively.

Two separate PCRs were set up to amplify the rearrangements of Vγ1 1-9–Vγ2 1-9–JGT1/2 and Vγ1 1-9–Vγ2 1-9–JGT3.

The amplified products were diluted at a ratio of 1:20 and prepared for denaturation with the standard ROX™ matrix for the subsequent analysis of the fragments using an ABI Genetic analyzer (Thermo Fisher Scientific, 168 Third Avenue, Waltham, MA, USA, 02451). The samples were then analyzed using Gene Mapper™ software v.6 (Thermo Fisher Scientific, 168 Third Avenue, Waltham, MA, USA, 02451). Each sample presented an electropherogram in a predetermined range between 270 and 230 bp that was the same for both rearrangements. Currently, 3 different results should be expected: (a) one or two peaks (monoallelic or biallelic clonality) within the expected length range and distant from the background in one or both rearrangements; the sample was positive for the presence of TCRγ clonal rearrangement with the presence of a clonal population. (b) The absence of one or two predominant peaks with the presence of a Gaussian distribution of the peaks always within the expected range; the sample was negative for the presence of a compatible TCRγ clonal rearrangement with the absence of a clonal population. (c) The presence of multiple predominant peaks; the sample was oligoclonal for the presence of an oligoclonal population.

For the detection of Ig heavy chain rearrangements, the “B-CELL Lymphoma Kit FL” (Experteam, Venice, Italy) was used through a semi nested PCR approach. In B lymphocytes, the variable region of the heavy chain consists of 3 framework regions in which there are conserved nucleotide sequences and 3 CDR regions in which there are hypervariable DNA sequences that code for the antigen-binding region and undergo the somatic hypermutation process.

The framework FR1, FR2, and FR3 segments were analyzed by means of three different reactions accumulated by a 3′ primer marked FAM, which recognizes the consensus region JH, while the 5′ primers recognize the conserved sequences of VH genes. The kit included a master mix to which 2 µL of DNA was added (50–100 ng/µL) and amplified by means of the corresponding protocols of the three segments. At this point, the semi nested PCR was set up using a second master mix and 1 µL of amplified product from the first reaction. The amplified products were prepared for denaturation with the standard ROX^TM^ matrix for the subsequent analysis of the fragments using an ABI Genetic analyzer (Thermo Fisher Scientific, 168 Third Avenue, Waltham, MA, USA, 02451).

The samples were then analyzed using Gene Mapper software (Thermo Fisher Scientific, 168 Third Avenue, Waltham, MA, USA, 02451). Each sample presented on an electropherogram in a predetermined range for the corresponding segments: FR1-JH between 340 and 408 bp, FR2-JH between 230 and 270 bp, and FR3-JH between 70 and 100 bp.

Again, 3 different results should be expected: (a) one or two predominant peaks (monoallelic or biallelic clonality) within the expected length range relating to either the FR1-Jh rearrangement, the FR2-JH rearrangement, or the FR3-JH rearrangement; the sample was positive for the presence of a clonal rearrangement of heavy chains compatible with the presence of a clonal population. (b) The absence of one or two predominant peaks with the presence of a Gaussian distribution of peaks always within the expected range; the sample was negative for the presence of a clonal rearrangement of heavy chains compatible with the absence of a clonal population. (c) The presence of more than two peaks (generally up to 5) within the expected range; the sample was positive for the presence of at least two clonal populations (oligoclonal) [36].

### 4.3. Statistical Analysis

Statistical analysis was performed using the Statistical Package OriginPro v2022 and R (version 4.3.1). Continuous variables were assessed for normality of distribution using the Kolmogorov–Smirnov test and expressed as the mean ± SD. Differences between groups were evaluated using analysis of variance (ANOVA), followed by Bonferroni’s correction for multiple comparisons. The application of parametric tests was justified by the central limit theorem considering the sample size of our study. Categorical variables were presented as frequencies or counts and compared using the chi-square test. A *p*-value < 0.05 was considered statistically significant. Correlations among the variables were examined using Spearman’s correlation coefficient, and correlation maps were used for visualization.

## 5. Conclusions

In conclusion, our study provides strong support for the hypothesis that patients with clinical symptoms resembling MC but without detectable cryoprecipitate may have circulating cryoglobulins. Our results highlight that the immune responses against specific HCV protein domains generate a range of biomarkers that are useful for assessing disease progression in HCV-positive patients. These biomarkers allow us to evaluate different phases of cryoglobulinemic syndrome, even in patients who test negative for cryoglobulins.

We have demonstrated significant similarities between patients with presumed CGs and those with type II CGs in terms of clinical symptoms; serological parameters (IgGs, FLCs, and RF); and clonal rearrangements of T/B-cell receptor genes.

In the era of precision medicine, there is a growing recognition of the importance of serological and molecular biomarkers in the clinical management of liver-related diseases [37]. In the context of extrahepatic HCV manifestations, such as MC, the identification of specific biomarkers holds great potential for guiding targeted therapies and personalized treatment strategies for patients.

Our findings suggest that the investigated laboratory parameters, as shown in Figure 4, Figure 5 and Figure 6 and Figure 8, have the potential to serve as effective biomarkers for enhancing the diagnosis, monitoring, and management of patients presenting with MC symptoms but without detectable cryoprecipitates. Furthermore, they offer valuable insights into the underlying mechanisms of MC. Further research is needed to validate and expand upon our findings, aiming to improve our understanding and clinical approach to mixed cryoglobulinemia.

## Figures and Tables

**Figure 1 ijms-24-11602-f001:**
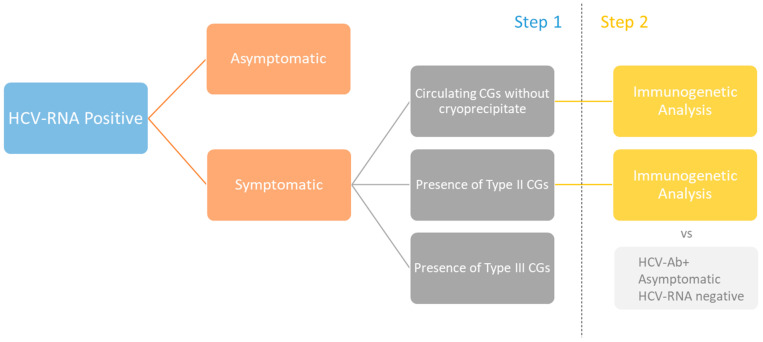
The steps of the analysis used in the study.

**Figure 2 ijms-24-11602-f002:**
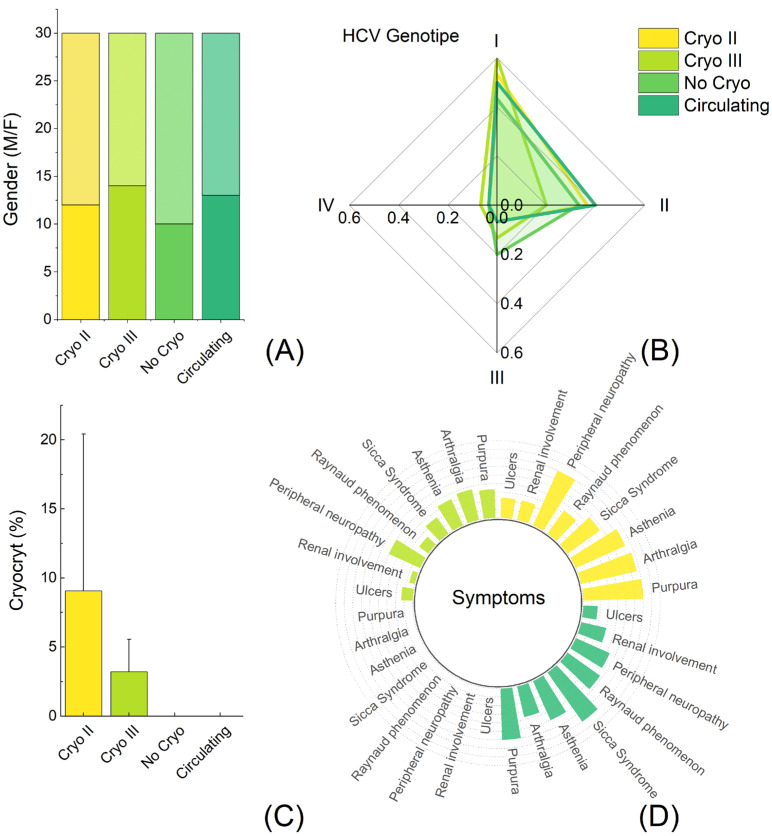
Graphical representation of the baseline characteristics of the 120 HCV-positive patients recruited in the study. Sex distribution (**A**), HCV genotype (**B**), cryocrit level (**C**), and cryoglobulin-related symptoms (**D**).

**Figure 3 ijms-24-11602-f003:**
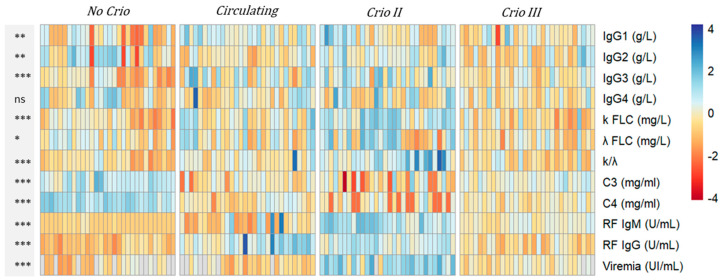
The heatmap displays the levels of the serological biomarkers in the 120 HCV-RNA-positive patients, categorized into four groups: No Cryo, (supposed) circulating Cryo, Cryo II, and Cryo III. The variables analyzed included IgG1 (g/L), IgG2 (g/L), IgG3 (g/L), IgG4 (g/L), k FLC (mg/L), λ FLC (mg/L), k/λ, C3 (mg/mL), C4 (mg/mL), RF IgM (U/mL), RF IgG (U/mL), and viremia (UI/mL). To aid in the comparison, the data underwent logarithmic transformation (with an offset of 1) and row-wise scaling. The color scale represents Z-scored values, with cooler colors indicating higher values and warmer colors indicating lower values. The color palette ranges from blue (representing higher values) to red (representing lower values). The gray band on the left side of the heatmap summarizes the *p*-values obtained from the ANOVA analysis for each parameter. Asterisks are used as follows: ns for not-significant, * for *p* < 0.05, ** for *p* < 0.01, and *** for highly significant *p*-values < 0.001. Gray cells indicate missing values.

**Figure 4 ijms-24-11602-f004:**
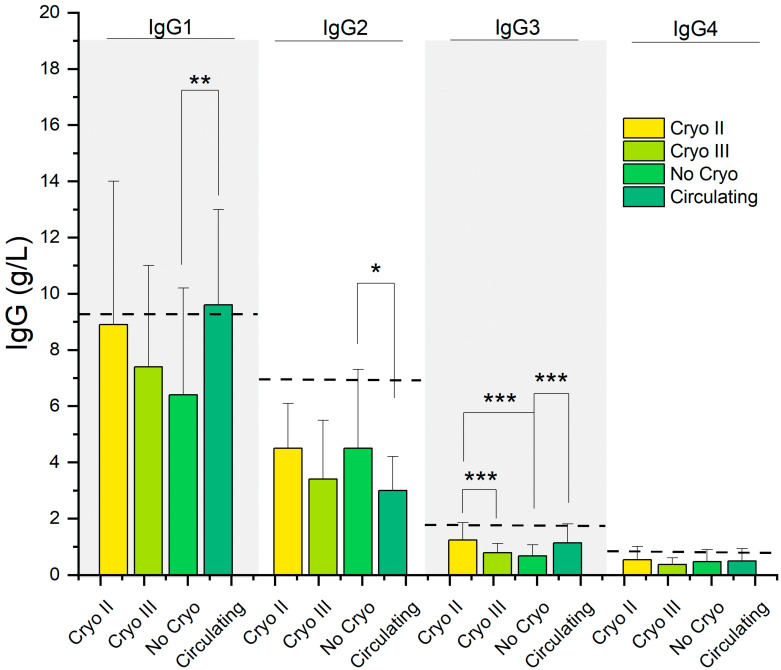
IgG subclasses in the different subgroups of the HCV-RNA-positive patients. *: *p* < 0.05; **: *p* < 0.01; ***: *p* < 0.001. Continuous dashed lines indicate the upper bound of the reference range for the serological parameter.

**Figure 5 ijms-24-11602-f005:**
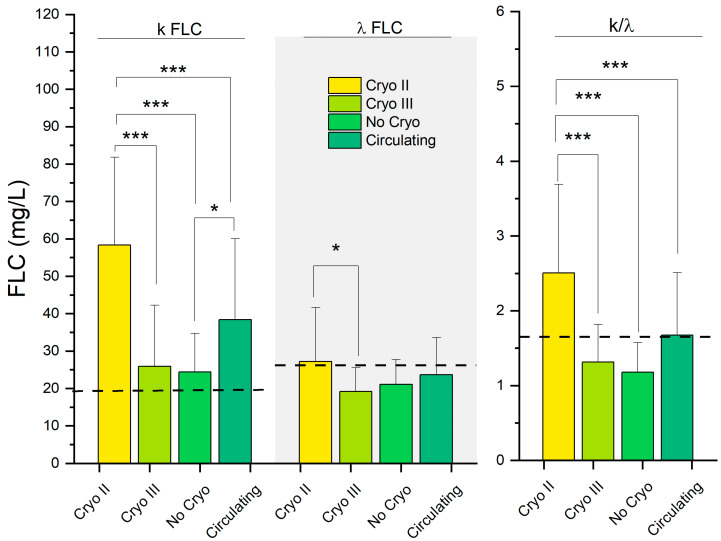
Serological FLC in the HCV-RNA-positive patients. *: *p* < 0.05; ***: *p* < 0.001. Continuous dashed lines indicate the upper bound of the reference range for the serological parameter.

**Figure 6 ijms-24-11602-f006:**
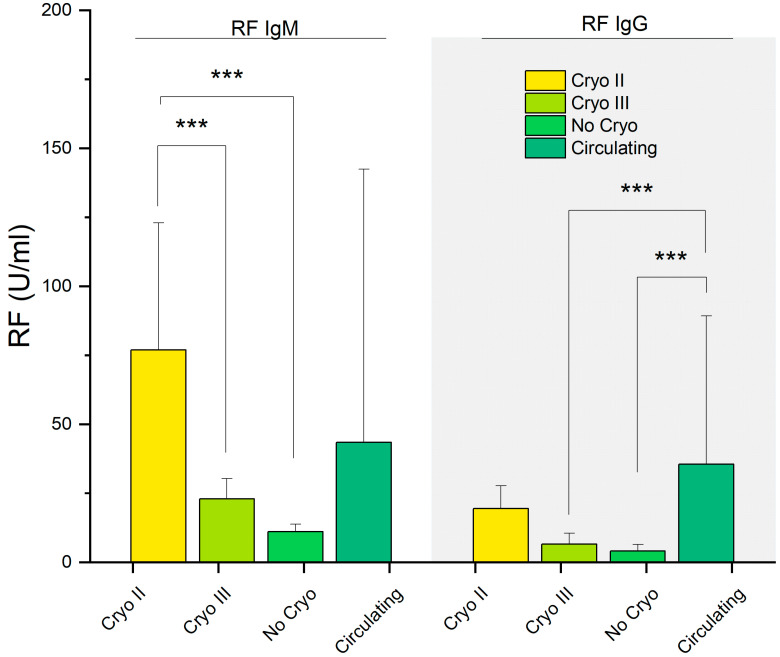
IgM and IgG levels of the rheumatoid factor in the HCV-RNA-positive patients. ***: *p* < 0.001.

**Figure 7 ijms-24-11602-f007:**
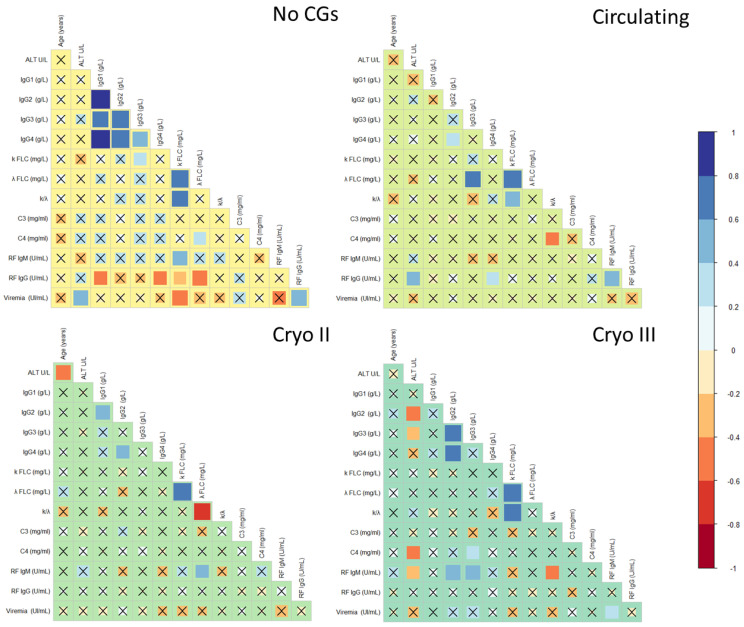
Correlation matrices of the immunological parameters measured in the four investigated groups, namely patients without CGs, patients with circulating CGs, and patients with type II and III CGs. The Pearson’s correlation coefficients are arranged in correlation matrices. No significant correlations are marked with an “×” symbol. Correlations are visualized using two scales simultaneously, a color scale and a size scale. The larger the square, the stronger the correlation. Dark blue square dots indicate a strong positive correlation, and dark red ones indicate a strong negative correlation.

**Figure 8 ijms-24-11602-f008:**
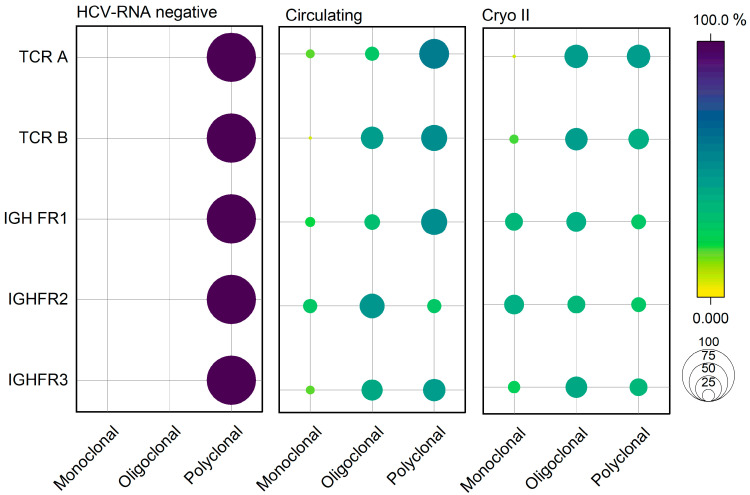
Clonal rearrangements for TCRγ and IGH FR1, FR2, and FR3 in patients who were recovered from HCV infection (HCV-RNA-negative with anti-HCV antibodies), in patients with possible circulating CGs, and in patients with type II CGs.

**Table 1 ijms-24-11602-t001:** Clinical and serological correlates of HCV-RNA-positive patients.

	**1° Group (30)**	**2° Group (30)**	**3° Group (30)**	**4° Group (30)**
Sex (M/F)	10/20	13/17	12/18	14/16
HCV-RNA	Yes	Yes	Yes	Yes
HCV genotype				
1 (%)	13	15	16	18
2 (%)	10	12	11	6
3 (%)	6	2	2	4
4 (%)	1	1	1	2
CG	No	Circulating	Type II	Type III
Cryocrit%	No	No	9.06 ± 11.36	3.21 ± 2.35
SYMPTOMS				
Purpura	No	18	21	10
Arthalgia	No	11	20	11
Asthenia	No	15	19	10
Sicca Syndrome	No	21	14	7
Raynaud phenomenon	No	14	10	4
Peripheral neuropathy	No	13	21	12
Renal involvment	No	9	7	2
Ulcers	No	5	7	4
Treatment	No	No	No	No

**Table 2 ijms-24-11602-t002:** Serological biomarkers of the 120 HCV-RNA-positive patients.

Features ^1^	Circulating *n* = 30	Cryo II *n* = 30	Cryo III *n* = 30	No Cryo *n* = 30	*p* ^2^	Circulating vs. Cryo II ^3^	Circulating vs. Cryo III ^3^	Cryo II vs. Cryo III ^3^	Circulating vs. No Cryo ^3^	Cryo II vs. No Cryo ^3^	Cryo III vs. No Cryo ^3^
ALT U/L	55.4 (27.5)	47.3 (37.3)	33.5 (14.9)	41.1 (22.7)	0.016	>0.9	0.012	0.3	0.2	>0.9	>0.9
IgG1 (g/L)	9.63 (3.40)	8.92 (5.12)	7.44 (3.57)	6.15 (3.76)	0.005	>0.9	0.2	>0.9	0.006	0.051	>0.9
IgG2 (g/L)	3.05 (1.22)	4.48 (1.93)	3.35 (2.11)	4.54 (2.80)	0.009	0.054	>0.9	0.2	0.041	>0.9	0.2
IgG3 (g/L)	1.16 (0.67)	1.23 (0.61)	0.79 (0.33)	0.58 (0.38)	<0.001	>0.9	0.073	0.008	<0.001	<0.001	0.7
IgG4 (g/L)	0.49 (0.45)	0.54 (0.47)	0.37 (0.24)	0.47 (0.43)	0.4	>0.9	>0.9	0.7	>0.9	>0.9	>0.9
k FLC (mg/L)	38.5 (21.7)	58.4 (23.5)	25.9 (16.3)	24.4 (10.4)	<0.001	<0.001	0.063	<0.001	0.025	<0.001	>0.9
λ FLC (mg/L)	23.7 (10.0)	27.2 (14.4)	19.2 (6.4)	21.1 (6.7)	0.014	>0.9	0.5	0.013	>0.9	0.11	>0.9
k/λ	1.67 (0.84)	2.51 (1.18)	1.32 (0.50)	1.18 (0.40)	<0.001	<0.001	0.5	<0.001	0.10	<0.001	>0.9
C3 (mg/mL)	94.0 (15.3)	89.4 (26.2)	104.7 (9.7)	119 (16)	<0.001	>0.9	0.13	0.007	<0.001	<0.001	0.013
C4 (mg/mL)	15.0 (4.2)	12.7 (6.5)	15.3 (3.1)	28.9 (5.8)	<0.001	0.5	>0.9	0.3	<0.001	<0.001	<0.001
RF IgM (U/mL)	43.4 (99.0)	76.9 (46.1)	22.9 (7.4)	11.1 (2.7)	<0.001	0.12	>0.9	0.001	0.15	<0.001	>0.9
RF IgG (U/mL)	35.5 (53.8)	19.5 (8.2)	6.6 (3.9)	4.13 (2.29)	<0.001	0.15	<0.001	0.4	<0.001	0.2	>0.9
Viremia (UI/mL)	25,011 (30,753)	3,429,521 (462,606)	55,999 (41,320)	34,729 (42,468)	<0.001	<0.001	>0.9	<0.001	>0.9	<0.001	>0.9

^1^ Mean (±SD), ^2^ one-way ANOVA, and ^3^ Bonferroni.

**Table 3 ijms-24-11602-t003:** Clonal rearrangements for TCRγ and IGH RF1, RF2, and RF3 in HCV-RNA-negative (neg) patients (who were recovered from infection, asymptomatic, without CGs (no Cryo), HCV-RNA-positive symptomatic patients with possible circulating CGs, and HCV-RNA-positive patients with type II CGs (Cryo II).

Features	Circulating, *n* = 30	Cryo II, *n* = 30	HCV-RNA Neg, *n* = 10	*p* ^1^	Circulating vs. Cryo II ^2^	Circulating vs. No Cryo ^2^	Cryo II vs. HCV-RNA Neg ^2^
TCRγ A				0.023	ns	ns	0.021
Monoclonal	17%	6.9%	0%				
Oligoclonal	27%	45%	0%				
Polyclonal	57%	48%	100%				
TCRγ B				0.010	ns	0.033	0.009
Monoclonal	6.7%	18%	0%				
Oligoclonal	43%	43%	0%				
Polyclonal	50%	39%	100%				
IGH FR1				0.003	ns	0.051	<0.001
Monoclonal	20%	34%	0%				
Oligoclonal	30%	38%	0%				
Polyclonal	50%	28%	100%				
IGHFR2				<0.001	ns	<0.001	<0.001
Monoclonal	27%	38%	0%				
Oligoclonal	47%	34%	0%				
Polyclonal	27%	28%	100%				
IGHFR3				0.009	ns	0.015	0.003
Monoclonal	17%	24%	0%				
Oligoclonal	40%	41%	0%				
Polyclonal	43%	34%	100%				

^1^ Fisher’s exact test and ^2^ Bonferroni; ns stands for not significant.

## Data Availability

The data presented in this study are available upon reasonable request from the corresponding author.

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
