# Peer review of "Serological and Molecular Characterization of Hepatitis C Virus-Related Cryoglobulinemic Vasculitis in Patients without Cryoprecipitate"

_ijms, 2023, doi:10.3390/ijms241411602_

Round 1

Reviewer 1 Report

The manuscript by Napodano et al. has merits but it needs to be extensively improved. It should be proofread by a native speaker. Here are just a few examples concerning the English that need attention:

line 60: residua --> residues;

line100: evidences --> evidence (it is an uncountable noun and is not used the plural);

Line 87: patients are aware hepatatis status... missing "OF THEIR" between aware andhepatitis;

Lines 417-420: this is completely uncomprehensible,

and so on.

The authors should include Materials and Methods section and present there with subtitles the patients, the sampling, the testing, the statistical analyses employed.

The "Results" section is poorly organized. Divide in different sestions to better comprehend the results.

In my opinion, the charts should to be replaced with dot plots which better represent  the variations in the different patients.

Table 1 must appear after its citation in the text. Check that issue also for the other figures.

Lines 96-97: give examples of such genes

Viremia is much higher in cryo II group (table 2). How does it correlate with the other parameters and groups?

The "Discussion" section must be improved (there is a multitude of little paragraphs, some of which are not at all clear, links are missing), and the conclusions should be clearly presented.

English needs to be improved.

Author Response

We gratefully thanks the Reviewer for the appreciated comment to our manus. We did our best to answer to all the queries and enclosed the changes suggested.

We corrected all the typos/errors pointed out by the Reviewer.

We better organized Materials and Methods as suggested.

As suggested by the Reviewer, we divided the "Results" section into four distinct thematic paragraphs. We would like to extend our appreciation for this valuable suggestion, as it has greatly enhanced the organization and comprehension of our findings.

We understand and acknowledge the Reviewer's concern regarding the necessity to display all the data points to investigate inter-subjects variability. To address this concern, we implemented a heatmap that includes all the measured parameters for each subject. This informative heatmap, presented as a new Figure 3 in our revised manuscript, allows for a comprehensive analysis of the data, and is discussed in conjunction with Table 2, which reports only means and standard deviation. As far as the following charts are concerned, taking also into consideration that all the data are provided in the new Figure 3, given the extensive number of parameters and groups analysed in each chart, we believe that utilizing a barplot would be a more suitable approach compared to a boxplot containing all data points. This choice allows for a more efficient utilization of the Figure space.

We checked that all Tables and Figures appear after their citation in the text.

We gave examples of genes (SNPs).

A statistically significant difference was observed for viremia between Cryo II and the other groups, but no correlation was observed with the other parameters.

We ameliorated Discussion as suggested, splitting it in subsections to better emphasize our results; we also added a final section of Conclusions.

Reviewer 2 Report

Thank you for this valuable research although some reviewing comments need clarification: 

in line 62 "If monoclonal cryoimmunoglobulinemia" need correction

in line 64 what is meant by nosographic?

in line 107 IGH abbreviation  how it refer to immunuglobulin receptor?

in introduction must give more details about type II and type III MC

in line 145 why  10 patients control group do not icluded in the 4 study group as fifth group?

in table 1 % of genotype not present and the treatment refer to which treatment of virus or treatment of manfestation of CG

Figure 2 need correction of C and D 

It is wrong to mention result in patients and methods

in Table 2 the unit of viremia  mention although not mention in patients and methods

 in figures 3 and followed figures the legand for astrex must be mentioned

in  line 304  different results about IgG4  as regard results mention in line 287

in figure 7 why yoyu do not include type III CG

in line 345 remove gamma from TCR

 in table 3 what is symbol @ refer to?

What is number 1 refer for what in table 3?

in  line 281  is contoversal to results in line 370  

in line 425 correct GC

Minor revision

Author Response

We gratefully thanks the Reviewer for the appreciated comments to our research.

We corrected all the words, typos, and abbreviations pointed out by the Reviewer.

We better detailed type II and type III MC in Introduction as suggested.

The control group of ten patients analysed in the second phase cannot be the fifth group of the first phase of analysis, because they are patients recovered from the infection, are HCV-RNA negative, and have not been analysed for the serological parameters analysed in the four groups of patients who were HCV-RNA positive. Therefore, we retain it is more appropriate to keep them separate.

We enrolled HCV positive patients characterized by the absence of antiviral treatment and/or immunosuppressive therapy or therapies for vasculitis (i.e., rituximab and plasma exchange cycles) (listed among the exclusion criteria).

We corrected Figure 2.

We mentioned viremia in Materials and methods as suggested.

We added legends for asterix as suggested in all the Figures.

In Table 3, number 1 is referred to Fisher's exact test (we added).

The addition of molecular analysis of TCR and IGH also for Type III represents an appreciable suggestion; but at this stage of evaluation, clinical features and serological results that we obtained from the first phase of analysis, pointed out a stronger similarity between Circulating and Cryo II (symptoms, IgG subclasses, k/λ ratio, RF IgM and IgG); for this reason we performed the molecular rearrangements only of Cryo II in comparison with Circulating and controls as we displayed in Figure 8 (the old Figure 7) .

Round 2

Reviewer 1 Report

The authors have well improved the text and the presentation of the manuscript and data.